# Clinical Utility of the Parent-Report Version of the Strengths and Difficulties Questionnaire (SDQ) in Latvian Child and Adolescent Psychiatry Practice

**DOI:** 10.3390/medicina58111599

**Published:** 2022-11-04

**Authors:** Ņikita Bezborodovs, Arta Kočāne, Elmārs Rancāns, Anita Villeruša

**Affiliations:** 1Department of Psychiatry and Narcology, Rīga Stradiņš University, LV-1007 Riga, Latvia; 2Child Psychiatry Clinic, Children’s Clinical University Hospital, LV-1004 Riga, Latvia; 3Department of Public Health and Epidemiology, Rīga Stradiņš University, LV-1007 Riga, Latvia

**Keywords:** children and adolescents, mental health, emotional and behavioural disorders, Strengths and Difficulties Questionnaire (SDQ)

## Abstract

*Background and Objectives*: Screening instruments can be crucial in child and adolescent mental healthcare practice by allowing professionals to triage the patient flow in a limited resource setting and help in clinical decision making. Our study aimed to examine whether the Strengths and Difficulties Questionnaire (SDQ), with the application of the original UK-based scoring algorithm, can reliably detect children and adolescents with different mental disorders in a clinical population sample. *Materials and Methods*: a total of 363 outpatients aged 2 to 17 years from two outpatient child psychiatry centres in Latvia were screened with the parent-report version of the SDQ and assigned clinical psychiatric diagnoses. The ability of the SDQ to predict the clinical diagnosis in major diagnostic groups (emotional, conduct, hyperactivity, and developmental disorders) was assessed. *Results*: The subscales of the parent-report SDQ showed a significant correlation with the corresponding clinical diagnoses. The sensitivity of the SDQ ranged 65–78%, and the specificity was 57–78%. The discriminative ability of the SDQ, as measured by the diagnostic odds ratio, did not quite reach the level of clinical utility in specialised psychiatric settings. *Conclusions*: We suggest the SDQ be used in primary healthcare settings, where it can be an essential tool to help family physicians recognise children needing further specialised psychiatric evaluation. There is a need to assess the psychometric properties and validate the SDQ in a larger populational sample in Latvia, determine the population-specific cut-off scores, and reassess the performance of the scale in primary healthcare practice.

## 1. Introduction

Mental, behavioural, and neurodevelopmental disorders in children and adolescents have been a rising concern during the last decades worldwide [1]. They have become the leading cause of disability in this population cohort in developed and developing countries alike [2].

Childhood and adolescence are critical stages of development for mental health and well-being throughout the lifespan. Most mental health problems of adulthood have their onset during or before adolescence [3] making this a critical period for recognition and treatment. Early identification and access to appropriate, evidence-based psychosocial interventions and support in childhood mental, behavioural, and neurodevelopmental disorders are essential to good recovery and better psychosocial functioning in adulthood [4,5].

While the global coverage of prevalence data for mental disorders in children and adolescents is limited, and only one-quarter of countries are collecting data on the number of children treated by a mental health professional [6], there is clear evidence of a massive gap between the number of children and adolescents needing mental health treatment and support and the number of children receiving it in the mental health services [7]. Mental health services worldwide are overwhelmed by demand that has been further increased as an effect of the COVID-19 pandemic, which can result in long waiting times and more suffering for the youngsters and families seeking help [8].

The introduction of screening procedures in mental health services can be a helpful step because it can potentially allow professionals to triage help-seeking patients based on their level of risk of having a mental disorder and determine the most appropriate treatment programme and level of care. This allows for youngsters with the highest risks and the highest need for intervention to be prioritised and for less time to be spent on psychiatric evaluation of healthy children. However, for a screening procedure to work, we must be sure that the screening tools used have reasonable validity and clinical utility in the population in which they are used [9].

The Strengths and Difficulties Questionnaire (SDQ) [10] has long been established as one of the most widely used screening instruments in child mental health research and clinical practice. It is easy to complete, relatively short, and user-friendly as it covers not only the child’s difficulties but also their strengths. It allows comparisons between different populations and is sensitive to change over time [11].

The SDQ consists of 25 items comprising of five subscales: emotional symptoms, peer relationship problems, conduct problems, hyperactivity/inattention, and pro-social behaviour. The “total difficulties” scale is a compound scale that can be calculated by summing up all the “difficulties” subscales, excluding the pro-social subscale that represents the child’s “strengths” [10]. In later studies, two other compound scales were proposed: the internalising difficulties scale, which is the sum of emotional problems and peer problems subscales, and an externalising problems scale which is the sum of conduct problems and hyperactivity subscales [12]. The SDQ is available in parent-report and teacher-report versions for 2- to 4-year-olds and 4- to 17-year-olds, as well as a self-report version for 11- to 17-year-olds.

Initially, the SDQ was suggested as a screening instrument in community screening programmes to potentially increase the detection of child psychiatric disorders, thereby improving access to effective treatments [13]. However, over the years, the SDQ has been used more and more in clinical settings as a measure of child psychopathology and in other types of research, e.g., aetiological, longitudinal, and service evaluation studies [11]. In the UK, parent and teacher versions of the SDQ demonstrated good validity, not only in discriminating between children with diagnosed mental disorders and a representative community sample [13] but also in identifying different categories of disorders within the clinical sample [14]. The parent and teacher SDQs proved to be valid and helpful questionnaires for use in the framework of a multi-dimensional behavioural assessment and appeared to be well suited for screening purposes, longitudinal monitoring of therapeutic effects, and scientific research purposes [15]. The findings by Hall et al. indicate that the SDQ is a valid outcome measure for use in RCTs and clinical settings [16].

There is a vast body of research on the internal consistency and reliability of the subscales of the SDQ in different contexts and populations with mixed results. In some studies, the SDQ exhibited strong internal consistency [17], but there are also studies expressing concerns regarding the reliability of the subscales, with most subscales showing only satisfactory or low internal consistency [18].

For younger age groups, it has been suggested that only the total difficulties score should be used for screening purposes because some subscales are unreliable [19]. In their systematic review, Kersten et al. suggested that an assessment of a pre-schooler should not rely on a single informant because of a moderate level of consistency between different informants [20].

Many studies have been done on the psychometric properties of the Strengths and Difficulties Questionnaire in different cultures. The Chinese version of the SDQ exhibited high levels of reliability and validity, indicating that the SDQ is appropriate for assessing psychopathology in Chinese adolescents [18]. The parent- and self-report versions of the SDQ showed good concurrent validity and psychometric properties in a Dutch community sample [21]. It was concluded that the results favour using the Swedish SDQ-S as a screening instrument for adolescents, despite the low internal consistencies of some of its subscales [22]. The usefulness of SDQ UK-based scoring algorithms in detecting mental health disorders among Norwegian patients was only partly supported; it seems best suited to identify children and adolescents who do not require further psychiatric evaluation [23].

The SDQ has been routinely used as a screening tool in Latvian child and adolescent mental health services because it is one of the rare screening tools that have been translated and adapted into the Latvian language in 2014 and is freely available for use in clinical practice from the developer’s website [24]. There is some data regarding the validity of the Latvian version of the SDQ in the populational samples, and it has been used in comparative research [25]. In a general school-based sample of 8–10-year-old children (*n* = 269), the Cronbach’s alfas for the Latvian parent-report SDQ subscales ranged from 0.49 to 0.70 [26]. In another recent study in a sample of 3–6-year-old children (*n* = 507), the reported internal consistency of the SDQ subscales ranged from 0.72 to 0.8 [27]. However, the validity and utility of the SDQ in the clinical sample of Latvian children and adolescents have never been evaluated. There is also no data available regarding the population-specific cut-off scores to differentiate normal and abnormal SDQ results, so in clinical practice, the original UK-based scoring algorithm is used in Latvia, which has been shown to be a potentially problematic practice in previous studies done in other countries [23].

This study aimed to examine whether the SDQ with the application of the original UK-based scoring algorithm, as it is currently used in the outpatient child-adolescent psychiatry settings in Latvia, can reliably detect children and adolescents with emotional disorders, conduct disorder, hyperactivity, or developmental disorders and to examine the sensitivity, specificity, and other predictive properties of this screening instrument in a sample of Latvian children and adolescents with clinically established mental health diagnoses.

## 2. Materials and Methods

### 2.1. Participants

The study sample consisted of 2–17-year-old children and adolescents who received outpatient psychiatric care in two outpatient psychiatry centres in Latvia from November 2019 to October 2020. Screening data were collected before the first-time psychiatric appointment from the patient’s parent in all participants of the study. In one of the study centres—Children’s Clinical University Hospital Child psychiatry clinic, located in the capital of Riga and providing secondary and tertiary psychiatric inpatient and outpatient care to paediatric patients from all over the country but mainly from the metropolitan region—the screening was a part of routine clinical practice, and the SDQ screening data were available for retrospective analysis in the patient’s medical documentation. In the other study centre—Hospital “Gintermuiza”, a specialised psychiatric hospital located in the city of Jelgava (Latvia’s 4th largest city) and providing secondary psychiatric inpatient and outpatient care to children and adults from mostly rural Zemgale region—the screening procedure was introduced for the purposes of this study, so written informed consent was obtained from the study participants before inclusion in the study.

The second centre was included in the study to make the clinical study sample, despite being a convenience sample, maximally representative of the types of outpatient child mental health services in Latvia and of the types of clinical help-seeking populations receiving care. There were no significant differences in the performance of the SDQ between centres in further analysis, so data on centre-specific characteristics is not separately reported in the Results section.

### 2.2. Questionnaires

The paper-based Latvian version of the parent-report SDQ was used. The questionnaire consists of 25 items which cover five subscales: emotional difficulties, peer problems, hyperactivity and inattention, conduct problems, and prosocial behaviour. Each item is rated on a 3-point scale (0 = not true, 1 = somewhat true, 2 = certainly true). Scores for each subscale were calculated according to the SDQ manual. The scale scores were calculated by summing the item scores per scale. Scales with at least one missing value were excluded from the analysis.

### 2.3. Diagnostic Algorithm

The positive screening result was defined using the UK-based scoring algorithm created by the author of the screening instrument based on a large UK community sample [18]. A positive screening result was defined as having “high” or “very high” level of difficulties, which in the original UK populational sample had identified children scoring higher than the 90th percentile. In previous research done in clinical samples both by Goodman in the UK population [14] and researchers in other countries [28], defining “caseness” using this algorithm has appeared to be more clinically relevant than choosing a lower (above 80th percentile) or higher (above 95th percentile) diagnostic threshold.

### 2.4. Clinical Diagnoses

Clinical psychiatric diagnoses were established by a board-certified child and adolescent psychiatrist with the involvement of other members of the multidisciplinary outpatient team (e.g., clinical psychologist), based on the thorough clinical-psychiatric investigation of the child and detailed anamnestic information from multiple informants. The psychiatric interview was performed with all study participants, and the clinician performing the diagnostic evaluation was blinded to the results of the screening test for the purposes of the study.

Clinical diagnoses were established and formulated according to the ICD-10 diagnostic guidelines. For the purposes of this study, the clinical diagnoses were further united in broader diagnostic groups according to the relevance of the subscales of the SDQ for capturing particular psychopathological phenomena, as reported by Goodman [13,14]. The category of “any emotional disorders” included any mood disorder (F3 category according to the ICD-10), neurotic, stress-related and somatoform disorders (F4), and emotional disorders with onset specific to childhood (F93). The category of “any conduct disorder” included conduct disorder (F91) and mixed disorders of conduct and emotions (F92). The “hyperkinetic disorder” category was defined identically to the corresponding ICD-10 category (F90). “Any conduct disorder” and “hyperkinetic disorder” groups were then merged to form a broader “externalising disorder” group. The category of “any developmental disability” included mental retardation (F7), mixed specific developmental disorder (F83), and pervasive developmental disorder (F84).

### 2.5. Statistical Analyses

Statistical analysis was performed with IBM SPSS Statistics, version 26 (IBM Corp., Armonk, NY, USA). Cronbach’s alfa coefficients were computed to evaluate the internal consistency of the SDQ subscales and compound scales. Chi-square analysis was used to establish the correlations between dichotomous screening results and clinical diagnoses. The screening efficiency statistics were calculated in terms of sensitivity, specificity, positive predictive value (PPV), negative predictive value (NPV), positive likelihood ratio (LHR^+^), negative likelihood ratio (LHR^−^), and diagnostic odds ratio (OR^D^). The confidence intervals for sensitivity and specificity estimates were calculated on the 95% level of confidence.

Sensitivity and specificity are important metrics of a screening test that describe the accuracy of the test. Still, for clinical practice, the more relevant metrics are PPV and NPV, which represent the probability that a positive or a negative screening result correctly reflects the presence or absence of a diagnosis. Likelihood ratios are summary statistics (ratios of probabilities) that show how a positive or negative screening result changes the likelihood of a patient being diagnosed with a particular disorder. OR^D^ is a summary measure that depicts the discriminative ability of a screening test. To interpret the results, we used the characteristics of the clinical usefulness of the test described by Fischer and colleagues, suggesting that for a test to have the potential to alter clinical decisions, it should have a LHR^+^ > 7 and a LHR^−^ < 0.3 or an OR^D^ > 20 [29].

## 3. Results

In total, 363 children were included in the study. Most of the study participants were male (*n* = 230, 63%). The mean age was 9.28 (SD = 3.82) years for males and 10.93 (SD = 4.11) years for females. Some 43.8% (*n* = 159) of the study group were 11–17 y.o. adolescents, 5–10 y.o. children constituted 46.3% (*n* = 168), and 9.9% (*n* = 36) were in 2–4 y.o. group.

Out of the 363 patients, 27.0% (*n* = 98) were diagnosed with an emotional disorder, 14.0% (*n* = 51) were diagnosed with conduct disorder, 22.6% (*n* = 82) were diagnosed with hyperkinetic disorder, and 26.2% (95) were diagnosed with a developmental disability. Also, 2.5% (*n* = 9) of patients did not receive a psychiatric diagnosis. However, 19.6% (*n* = 71) were diagnosed with some other mental disorder (e.g., organic mental disorder, eating disorder, tic disorder, etc.).

A total of 314 (86.5%) patients had received clinical diagnoses falling into only one of the categories mentioned above. Of these, 38 patients (10.5%) had two comorbid diagnoses falling into different diagnostic groups, one patient (0.3%) had 3, and 1 patient (0.3%) had received comorbid diagnoses in 4 different groups.

The internal consistency statistics of the five parent-report SDQ subscales and compound scales are presented in Table 1. Emotional problems, hyperactivity, and prosocial subscales, as well as the externalising and total difficulties scales, demonstrated acceptable internal consistency (Cronbach’s alfa > 0.7). The results for the conduct problems and internalising difficulties scales were also close to being on the acceptable level. The peer problems subscale was the only SDQ scale with poor internal consistency.

The results of the parent-report SDQ screening are summarised in Table 2. As expected in a clinical help-seeking sample, the screening results show much higher degrees of reported psychopathology than is usually found in the populational cohorts. The positive screening results for different SDQ subscales ranged from 31.8% to 60.2%, with 60.9% of patients screening positive on the total difficulties scale. The dichotomous screening results for the compound internalising and externalising subscales could not be calculated because Goodman reported no cut-off values for the original UK sample.

Table 3 presents the screening efficiency of the parent-report version of the SDQ. A diagnosis of any emotional disorder significantly correlated with a positive screening result in emotional problems and a negative screening result in peer problems and hyperactivity. Any conduct disorder, hyperkinetic disorder, and externalising disorder significantly correlated with a positive screening result in hyperactivity, conduct problems, and total difficulties. Hyperkinetic disorder also had a correlation with a negative screening result in emotional problems. Developmental disability significantly correlated with a positive screening result in peer problems and low or very low result on the prosocial scale.

The sensitivity and specificity of appropriate subscales of the parent-report SDQ were 67%, CI (0.57, 0.77) and 57%, CI (0.50, 0.64) for any emotional disorder; 78%, CI (0.67, 0.89) and 57%, CI (0.50, 0.64) for any conduct disorder; 65%, CI (0.55, 0.75) and 78%, CI (0.73, 0.83) for the hyperkinetic disorder; and 72%, CI (0.63, 0.81) and 44%, CI (0.36, 0.52) for developmental disability.

Overall, none of the subscales of the SDQ has reached the interval of potential usefulness for clinical decision-making, based on the LHR^+^, LHR^−^, and OR^D^ results (LHR^+^ > 7, LHR^−^ < 0.3, OR^D^ > 20).

For example, the positive screening results in the parent-report SDQ hyperactivity scale had a LHR+ of 2.95, which means that the likelihood of a child getting diagnosed with hyperkinetic disorder after a positive screening test was 2.95 times higher, whereas after getting a negative screening result, the possibility of getting a clinical diagnosis was 0.45 times lower (LHR^−^). The aggregated chances of getting diagnosed with hyperkinetic disorder after a positive screening test were 6.56 times higher (OR^D^) than after a negative screening test, which is interpreted as not a significant enough difference to substantially influence clinical decisions in this highly saturated patient population [28,29].

## 4. Discussion

The use of screening procedures is a widespread and potentially beneficial practice in mental health services worldwide [9]. The benefits of screening might be even greater in lower resource settings as it can allow making use of the limited resources more effectively by triaging the patient flow [30]. However, as is the case in other areas of medicine, for the screening to be beneficial, we must be sure that the instruments we use are “fit for purpose”. Unfortunately, most of the research regarding the predictive properties of available mental health screening instruments in real-life clinical practice is done in high-resource settings and big countries where these instruments are usually developed, and there is little evidence regarding the clinical utility of these instruments in lower resource and smaller country conditions [28].

The primary purpose of our study was to evaluate the clinical utility of the parent-report version of the SDQ as it is currently used in Latvian child and adolescent mental healthcare settings, with the application of the original UK-based scoring algorithm developed by the author of the screening instrument.

The results of our study regarding the level of psychopathology detected by the SDQ using the same UK-based scoring algorithm in a clinical child and adolescent sample appear to be comparable to clinical samples from other countries. The number of Latvian children scoring positive in parent-report SDQ for emotional difficulties (49%) was higher than reported in Norway (25%) and the UK (33%) but slightly lower than in Bangladesh (55%). In contrast, in conduct problems, our results (47.9%) were significantly lower than in the UK (70%) but slightly higher than in Norway (40%) or Bangladesh (34%). In hyperactivity, Latvian results (31.8%) were similar to Norway (30%), slightly lower than Bangladesh (37%), and significantly lower than the UK (46%) [14,28].

The cross-cultural differences in the level of psychopathology detected by the parent-report SDQ can likely be explained by the differences in the psychometric functioning of the screening instrument in different cultural settings (cross-cultural validity of the SDQ in the absence of proper psychometric data and country-specific norms cannot be safely assumed [30,31,32]), and the differences in the functioning of the local mental healthcare systems (e.g., possible selection bias in the process of forming the clinical sample in the context of a specific mental healthcare system), rather than the actual differences in the levels of psychopathology in the population.

In our clinical sample, all the SDQ scales and subscales showed reasonable internal consistency, except the peer problem subscale which demonstrated poor internal consistency. In their systematic review of the psychometric properties of the SDQ, Kersten et al. have synthesised internal consistency estimates from 26 studies and found that the weighted average Chronbach’s alfa for the peer problem subscale of the parent-report SDQ was 0.49, for conduct problems, 0.56, and for other subscales in the range of 0.62 to 0.69, but for the total difficulties scale, 0.76, which is a pattern similar to the one found in our clinical data [20]. Interestingly, a similar pattern of internal consistencies was recently reported for the adolescent self-report version of the SDQ. In a comparative study of datasets from seven different countries (Bulgaria, Germany, Greece, Netherlands, Poland, Romania, Slovenia), Duinhof et al. report the Chronbach’s alfas for the peer problem subscale ranging from 0.55 to 0.65, which was lower than for other subscales of the SDQ that all reached acceptable levels of internal consistency (above 0.7) [31]. These findings can indicate that the factor structure of the parent-report version of the SDQ in non-UK populations might differ from the original factor structure of the screening instrument, with peer problem subscale items being the most problematic for cross-cultural application.

The sensitivity, specificity, and other predictive properties of the parent-report version of the SDQ found in the Latvian clinical sample are similar to the performance estimates of this screening instrument with the application of the original UK-based scoring algorithm in clinical samples of children and adolescents in other countries of the world [14,33]. In the study done by Goodman et al. in a looked-after child population in the UK, they found a sensitivity of 60.4% for conduct disorder, 78.6% for hyperkinetic disorder, and 64.3% for any anxiety or depressive disorder [34], which is similar to our findings and higher than the sensitivity and specificity reported by Goodman in a single-informant screening in the community sample [14]. In a study by Brøndbo et al. done in the Norwegian child mental healthcare setting, the predictive properties of the parent-report version of the SDQ for conduct disorder, hyperactivity, and emotional disorders were as follows: sensitivity of 0.83, 0.77, 0.47; specificity of 0.75, 0.80, 0.87; PPV of 0.59, 0.46, 0.66; NPV of 0.91, 0.94, 0.76; LHR^+^ of 3.29, 3.91, 3.68; LHR^−^ of 0.23, 0.29, 0.61; OR^D^ of 14.41, 13.35, 6.05, respectively [23]. These findings are somewhat better than the ones in our sample. Still, similarly to our findings, in Norway, the single informant parent-report version of the SDQ did not reach a level of clinical utility that could be deemed sufficient and warrant its use as a screening tool in a highly psychopathologically saturated clinical sample of children and adolescents.

### Strengths and Limitations

Despite the SDQ being a highly studied and widely used screening instrument both in scientific research and clinical practice, this is one of the few studies examining not only the psychometric and predictive properties of the scale in a particular population, but also its clinical utility as used in real-life practice. The clinicians that have established the clinical diagnoses were blinded to the results of the screening procedure to avoid biasing the result toward better agreement between the results of the screening and the clinical diagnoses. The study has been done in a sufficiently large group of patients to make inferences regarding the scale’s psychometric properties with a subject-to-item ratio of 14.5.

There are a number of limitations to this study. The study sample was a convenience sample of first-time psychiatric outpatients, so there is a potential for selection bias, although the patient population included in the study could be regarded as representative of Latvian clinical day-to-day practice and is ecologically valid. Another major limitation is the use of a clinical psychiatric diagnosis as the study’s assumed “golden standard”. For example, in our sample, the hyperactivity and conduct problems subscales of the SDQ appeared to be very similar in their ability to predict the diagnosis of both hyperkinetic disorder and conduct disorder, which rather than being just a problem of the discriminant ability of the subscales is also likely to indicate that in Latvian clinical child psychiatry, practice physicians tend to use these clinical psychiatric diagnoses interchangeably and not discriminate between conduct problems with and without an underlying attention deficit hyperactivity disorder neurophenotype. The use of the UK-based scoring algorithm to define “positive” screening results, although being the only valid option in the absence of local population-based norms, can also be regarded as a limiting factor, as well as the use of a single-informant screening protocol. Both the predictive properties and the clinical utility of the SDQ could potentially be improved by employing a multi-informant screening protocol (based on the combination of parent-, teacher-, and self-report SDQ) as suggested by the author of the screening instrument [13,14,34].

## 5. Conclusions

This study is the first examination of the Latvian parent-report version of the SDQ performed in a clinical population sample to date. Its findings have the potential to influence the way this screening instrument is used in Latvian clinical practice. Our findings illustrate the need to assess not only the psychometric properties of a scale but also its clinical utility when making a decision to introduce a screening procedure to clinical practice. The findings of this study also add to the ongoing discussion on the cross-cultural applicability of this psychometric instrument [23,31,32].

Our study suggests that the parent-report version of the SDQ, as it is currently used in Latvian child and adolescent mental healthcare practice, has sufficient internal consistency, and the relevant subscales of the SDQ significantly correlate with the clinical diagnoses of emotional disorders, conduct disorder, hyperactivity, and developmental disability in a clinical sample.

However, the predictive properties and performance of the scale in the Latvian clinical population suggest that it might be more suitable for use in less clinically saturated samples, e.g., in populational research or as a screening tool to be used by family physicians in the primary healthcare settings to assist the decision of whether to refer the child to a specialised child mental health service for further evaluation.

Based on the results of our analysis, we suggest using an aggregated externalising difficulties score to screen for any externalising disorder rather than using hyperactivity or conduct problem scores separately.

The performance of the parent-report SDQ can be further improved by formulating the country-specific normative thresholds based on populational data in a larger general-population cohort, further investigating the factor structure and psychometric properties of the screening instrument and possibly assessing its predictive properties in a primary healthcare setting.

## Figures and Tables

**Table 1 medicina-58-01599-t001:** Internal consistency statistics (Cronbach’s alfas) of the parent-report version of the SDQ in a clinical population of Latvian children and adolescents.

SDQ Scale	Cronbach’s Alfa
Emotional problems	0.717
Peer problems	0.566
Hyperactivity	0.768
Conduct problems	0.676
Prosocial	0.777
Internalising difficulties	0.691
Externalising difficulties	0.810
Total difficulties	0.786

**Table 2 medicina-58-01599-t002:** Results of the parent-report SDQ screening in a clinical sample of Latvian children and adolescents.

SDQ Scale	N of Patients	Positive Screens	%
Emotional problems	353	174	49.3%
Peer problems	344	207	60.2%
Hyperactivity	352	112	31.8%
Conduct problems	351	168	47.9%
Non prosocial	358	147	41.1%
Total difficulties	327	199	60.9%

**Table 3 medicina-58-01599-t003:** Screening efficiency of the parent-report SDQ version subscales for different mental health diagnostic categories in a clinical sample of Latvian children and adolescents.

SDQ Parent-Report	N	TN	%	TP	%	FN	%	FP	%	Sig	Sen	Spe	PPV	NPV	LHR^+^	LHR^−^	OR^D^
Any emotional disorder																	
Emotional problems	353	149	42.2%	62	17.6%	30	8.5%	112	31.7%	0.000 **	0.67	0.57	0.36	0.83	1.57	0.57	2.75
Peer problems	344	93	27.0%	46	13.4%	44	12.8%	161	46.8%	0.041 *	0.51	0.37	0.22	0.68	0.81	1.34	0.6
Hyperactivity	352	159	45.2%	13	3.7%	81	23.0%	99	28.1%	0.000 **	0.14	0.62	0.12	0.66	0.36	1.4	0.26
Conduct problems	351	127	36.2%	38	10.8%	56	16.0%	130	37.0%	0.092	0.4	0.49	0.23	0.69	0.8	1.21	0.66
Non prosocial	358	147	41.1%	31	8.7%	64	17.9%	116	32.4%	0.051	0.33	0.56	0.21	0.7	0.74	1.21	0.61
Total difficulties	327	93	28.4%	49	15.0%	35	10.7%	150	45.9%	0.583	0.58	0.38	0.25	0.73	0.95	1.09	0.87
Any conduct disorder																	
Emotional problems	353	153	43.3%	23	6.5%	26	7.4%	151	42.8%	0.723	0.47	0.5	0.13	0.85	0.94	1.05	0.9
Peer problems	344	121	35.2%	34	9.9%	16	4.7%	173	50.3%	0.221	0.68	0.41	0.16	0.88	1.16	0.78	1.49
Hyperactivity	352	215	61.1%	24	6.8%	25	7.1%	88	25.0%	0.005 **	0.49	0.71	0.21	0.9	1.69	0.72	2.35
Conduct problems	351	172	49.0%	40	11.4%	11	3.1%	128	36.5%	0.000 **	0.78	0.57	0.24	0.94	1.84	0.38	4.89
Non prosocial	358	184	51.4%	24	6.7%	27	7.5%	123	34.4%	0.347	0.47	0.6	0.16	0.87	1.17	0.88	1.33
Total difficulties	327	117	35.8%	36	11.0%	11	3.4%	163	49.8%	0.017 *	0.77	0.42	0.18	0.91	1.32	0.56	2.35
Hyperkinetic disorder																	
Emotional problems	353	127	36.0%	28	7.9%	52	14.7%	146	41.4%	0.004 **	0.35	0.47	0.16	0.71	0.65	1.4	0.47
Peer problems	344	110	32.0%	50	14.5%	27	7.8%	157	45.6%	0.333	0.65	0.41	0.24	0.8	1.1	0.85	1.3
Hyperactivity	352	212	60.2%	52	14.8%	28	8.0%	60	17.0%	0.000 **	0.65	0.78	0.46	0.88	2.95	0.45	6.56
Conduct problems	351	159	45.3%	56	16.0%	24	6.8%	112	31.9%	0.000 **	0.7	0.59	0.33	0.87	1.69	0.51	3.31
Non prosocial	358	161	45.0%	31	8.7%	50	14.0%	116	32.4%	0.562	0.38	0.58	0.21	0.76	0.91	1.06	0.86
Total difficulties	327	112	34.3%	56	17.1%	16	4.9%	143	43.7%	0.001 **	0.78	0.44	0.28	0.88	1.39	0.51	2.74
Developmental disability																	
Emotional problems	353	134	38.0%	48	13.6%	45	12.7%	126	35.7%	0.602	0.52	0.52	0.28	0.75	1.07	0.94	1.13
Peer problems	344	111	32.3%	67	19.5%	26	7.6%	140	40.7%	0.006 **	0.72	0.44	0.32	0.81	1.29	0.63	2.04
Hyperactivity	352	178	50.6%	31	8.8%	62	17.6%	81	23.0%	0.715	0.33	0.69	0.28	0.74	1.07	0.97	1.1
Conduct problems	351	134	38.2%	42	12.0%	49	14.0%	126	35.9%	0.704	0.46	0.52	0.25	0.73	0.95	1.04	0.91
Non prosocial	358	171	47.8%	55	15.4%	40	11.2%	92	25.7%	0.000 **	0.58	0.65	0.37	0.81	1.66	0.65	2.56
Total difficulties	327	100	30.6%	60	18.3%	28	8.6%	139	42.5%	0.100	0.68	0.42	0.3	0.78	1.17	0.76	1.54
Externalising disorder																	
Emotional problems	353	109	0.309	50	0.142	70	0.198	125	0.354	0.040 *	0.42	0.47	0.29	0.61	0.78	1.25	0.62
Peer problems	344	98	0.285	78	0.227	39	0.113	129	0.375	0.077	0.67	0.43	0.38	0.72	1.17	0.77	1.52
Hyperactivity	352	190	0.54	69	0.196	50	0.142	43	0.122	0.000 **	0.58	0.82	0.62	0.79	3.14	0.52	6.10
Conduct problems	351	148	0.422	86	0.245	35	0.1	82	0.234	0.000 **	0.71	0.64	0.51	0.81	1.99	0.45	4.43
Prosocial	358	140	0.391	51	0.142	71	0.198	96	0.268	0.837	0.42	0.59	0.35	0.66	1.03	0.98	1.05
Total difficulties	327	103	0.315	85	0.26	25	0.076	114	0.349	0.000 **	0.77	0.47	0.43	0.80	1.47	0.48	3.07

N—total number of patients included in the analysis; TN—true negative screening results; TP—true positive screening results; FN—false negative screening results; FP—false positive screening results; Sig—Pearson Chi-square asymptotic significance (2-sided); Sen—Sensitivity (TP/(TP + FN); Spe—Specificity (TN/(TN + FP); PPV-Positive predictive value (TP/(TP + FP); NPV Negative predictive value (TN/(TN + FN); LHR^+^-Positive likelihood ratio (Sen/(1-Spe)); LHR^−^-Negative likelihood ratio ((1-Sen)/Spe); OR^D^-Diagnostic odds ratio (LHR^+^/LHR^−^); * *p* < 0.05; ** *p* < 0.01.

## Data Availability

The full study protocol and datasets used and/or analysed during the current study are available from the corresponding author upon reasonable request.

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
