# Peer review of "Clinical Utility of the Parent-Report Version of the Strengths and Difficulties Questionnaire (SDQ) in Latvian Child and Adolescent Psychiatry Practice"

_medicina, 2022, doi:10.3390/medicina58111599_

Round 1

Reviewer 1 Report

The purpose of the study was to examine clinical utility of a screening test (the Latvian parent-report version of the Strengths and Difficulties Questionnaire) using clinical psychiatric diagnosis as the reference standard.

1. Materials and Methods:

*Please mention psychometric properties of the Latvian version of SDQ.

2. Materials and Methods: Line 172: "Clinical diagnoses were formulated according to the ICD-10 diagnostic guidelines."

*Did “the child and adolescent psychiatrists” evaluate and diagnose based on ICD criteria?

3. Materials and Methods:

*Was the person performing/interpreting the screening test blind to the results of the reference standard and vice versa? *Were both the screening and the reference standard applied blind?

4. Materials and Methods:

*How was the study sample size determined?

5. Materials and Methods:

*Please make it clear in the paper whether the results of the test influenced the decision to perform the reference standard. For example, the reference standard was applied only to those with positive screening results? Or were all study participants both evaluated by a psychiatrist and screened?

6. Materials and Methods:

*Please describe the method for performing the test and the reference standard in sufficient detail.

7. Results:

*Please present sensitivities and specificities with confidence intervals.

8. Results: Line 253: "… which is comparable to the screening performance results of the parent-report SDQ found in other clinical populations previously."

*It seems that the sentence should be mentioned in the "Discussion" section.

9. Results: Lines 255-257: "Overall, none of the subscales of the SDQ has reached the interval of potential usefulness for clinical decision making, based on the LHR+, LHR- and ORD results (LHR+ > 7, LHR- < 0,3, ORD > 20)."

*Please mention the criteria in the "Materials and Methods" section.

10. Results:

*Was there a relationship between clinical utility of the parent-report version of the Strengths and Difficulties Questionnaire and children's age, parents' education level and being an only child?

In other words, was clinical utility of SDQ worse for children with younger ages, parents with less education, and parents with less parenting experience?

11. *Please attach the English and the Latvian versions of the SDQ to the article.

12. Discussion:

*Please, in the discussion section, compare the study results with the results of other studies in more detail regarding sensitivity, specificity, positive predictive value (PPV), negative predictive value (NPV), positive likelihood ratio (LHR+), negative likelihood ratio (LHR-) and diagnostic odds ratio (ORD).

13. *Please mention where can the full study protocol be accessed?

Reviewer 2 Report

A literature review section is required. There is a need of structuring the discussion to ensure that the methodological aspects are clearly presented. The manuscript will benefit from further discussion of key concepts and methodological criteria in order to offer a better articulation between theory and data. Data gathering and data analysis can be reconsidered and discussed more comprehensively. You should compare your results with others in terms of concrete data for better research integrative value. Please provide more details regarding the study limitations and strengths and what this means for the study findings. The conclusion, too short, should clarify the main contribution of the paper and the value added to the field. The proportion of old references is too high.

Round 2

Reviewer 1 Report

The paper can be accepted in present form.

Reviewer 2 Report

This revised version can be published.